# Is Osteopontin a Good Marker for Bone Metastasis in Canine Mammary Gland Tumor and Prostate Cancer?

**DOI:** 10.3390/ani13203211

**Published:** 2023-10-14

**Authors:** Caroline Grisoni Sanchez, Marxa Leão Figueiredo, Laíza de Sartori Camargo, Luiz Guilherme Dercore Benevenuto, Zara Alves Lacerda, Carlos Eduardo Fonseca-Alves

**Affiliations:** 1Institute of Health Sciences, Paulista University-UNIP, Bauru 17048-290, Brazil; karolzinha_sanchez@hotmail.com; 2College of Veterinary Medicine, Purdue University, West Lafayette, IN 47907, USA; mlfiguei@purdue.edu; 3School of Veterinary Medicine and Animal Science, São Paulo State University—UNESP, Botucatu 18610-160, Brazil; laiza.camargo@unesp.br (L.d.S.C.); l.benevenuto@unesp.br (L.G.D.B.); zara.lacerda@unesp.br (Z.A.L.); 4Veterinary Oncology Clinic—SEOVET, São Paulo 05016-000, Brazil

**Keywords:** canine, carcinoma, osteopontin, mammary gland, prostate, metastasis

## Abstract

**Simple Summary:**

Bone metastasis can develop from several tumor subtypes, and the presence of bone metastasis confers a negative prognosis to patients. In dogs, a small group of tumors, including urogenital and mammary gland tumors (MGTs), is known to develop bone metastases. The mechanisms involved in the development of bone metastases in dogs are unknown, and few studies have been published on this subject. Osteopontin (OPN) is a glycoprotein involved in tumor progression, angiogenesis, and metastasis. Several studies have implicated OPN overexpression in a higher incidence of bone metastases. In addition, OPN overexpression has been shown to be correlated with increased bone resorption in patients with cancer. Although OPN expression has been shown to be important in several cancer subtypes, in Veterinary Medicine, no previous studies had investigated the role of OPN in patients with bone metastasis. Therefore, our study aimed to evaluate OPN levels immunohistochemically and associate them with the detection of bone metastasis in canine mammary tumors and prostate cancer.

**Abstract:**

Osteopontin (OPN) is a protein synthesized by a large number of cells, and its overexpression has been associated with the development and prognosis of cancer. OPN overexpression has been claimed to be a marker for the development of bone metastasis in human cancers, but no prior research has investigated the association between OPN expression and the metastasis of canine mammary gland tumors (MGTs) and prostate cancer (PC). Therefore, we investigated OPN expression in MGTs and PC samples from 50 canine patients with or without metastasis (bone vs. other sites). Higher OPN expression was detected in primary tumor samples from animals with bone metastasis than in those without bone involvement (*p* = 0.0321). In MGT samples, a significantly lower survival rate was observed in patients with higher OPN expression (*p* = 0.0171). In animals with PC, there was a strong trend toward lower survival in animals with positive OPN expression; however, this trend was not statistically significant (*p* = 0.0779). From these findings, it can be concluded that OPN may be a promising target for future MGTs and PC studies because of its role in enhancing cell invasion and metastasis.

## 1. Introduction

Metastatic spread is a complex process that involves several main steps such as neovascularization, local invasion, and subsequent intravasation, transportation, and obstruction of vessels [1,2]. These coordinated events are guided by molecular, cellular, and biochemical alterations, leading to tumor cell invasion, migration, and the establishment of a new neoplastic site [2,3]. Among the metastatic sites, bone metastasis is a complex and poorly understood process in Veterinary Medicine and is often associated with poor prognosis. The development of effective antimetastatic drugs requires an understanding of the molecular mechanisms driving metastasis, differences in tumor cells colonizing distant organs, and the recognition of potential molecular targets [1,2].

Bone metastasis is a multifaceted process which includes tissue matrix remodeling and disturbance of the local immune system, endothelial cells, fibroblasts, osteoblasts, and osteoclasts [1,2,3]. Among the few studies focusing on the role of molecules in this process, osteopontin (OPN), also known as secreted phosphoprotein 1, it is implicated in the control and regulation of local inflammation and immunity, and it is linked with metastatic cancer prognosis and overall survival [4,5,6,7]. In addition, OPN plays a role in tumor growth as a paracrine and autocrine mediator produced by macrophages and fibroblasts, leading to tumor invasion and angiogenesis, both of which correlate with poor prognosis [8].

In the pathogenesis of metastasis, OPN promotes the migration and invasion of cancer cells, allowing them to pass through the blood or lymphatic vessels and enter the blood or lymph stream. Additionally, OPN may facilitate the adhesion of cancer cells to distant tissues, promoting the formation of metastatic focus [4,5,6]. This glycoprotein is a member of a family of small integrin-binding proteins that have been strongly implicated in the development of bone metastasis [9,10]. Tissue and serum expression have been the focus of investigation in several cancer subtypes related to bone metastasis development and are associated with a more aggressive phenotype and poorer prognosis. Its actions play an essential role in cell–cell and cell–matrix communication, in addition to shaping cell behavior through paracrine mechanisms and autocrine processes [11]. OPN expression is controlled by several signaling molecules, such as epidermal growth factor (EGF), transforming growth factor beta (TGF-β), tumor necrosis factor β (TNFβ), interferon gamma (IFN-β), and interleukin-1β (IL-1β) [12].

OPN also contributes to a favorable environment for cancer cell growth in metastatic bone regions, acting in the degradation of bone tissue through osteoclast interactions, which promotes a favorable environment for cancer cells to invade the metastatic site [5]. It can interact with cells of the immune system and suppress immune responses that would normally destroy cancer cells. In addition, OPN positively influences angiogenesis by providing nutrients and oxygen to growing cancer cells [8]. Because of its involvement in the pathogenesis of metastasis, targeting OPN may serve as a promising therapeutic strategy for treating various cancers. However, further studies are needed to fully understand the mechanisms by which OPN promotes cancer progression and develop effective OPN-targeting therapies [5].

Little is known about the proteins and molecules involved in the pathogenesis of bone metastasis in Veterinary Medicine. Tumors with potential bone metastases are models for understanding this process. Among them, mammary gland tumors (MGTs) and prostate cancer (PC) in dogs are the most associated with bone metastasis and represent a comparative model for humans because of their similar clinical and histopathological characteristics [13].

PC is one of the most common cancers in men, and OPN has been implicated as a biomarker of human PC [7]. Moreover, studies have linked OPN plasma expression levels to the development of lung, prostate, and breast cancers [14,15,16], suggesting that OPN plays a role in the development and progression of these diseases. In human neuroendocrine tumors, OPN is overexpressed in patient serum, and during treatment, higher OPN levels (above 200 ng/mL) at initial evaluation predict a worse prognosis, leading to a shorter progression-free survival [17]. Therefore, owing to its high expression, OPN has been proposed as a tissue and serum marker for detecting bone metastasis and to help determine prognosis in humans, making it a promising therapeutic strategy for patients with PC [18]. 

Regarding canine PC, no previous study has investigated OPN expression in prostate tissues. Other than humans, dogs are the only species that spontaneously develop benign prostatic hyperplasia (BPH) and PC, whereas dogs with or without invasive cancer can develop benign prostatic intraepithelial neoplasia (PIN). However, regarding the OPN expression, no previous studies have reported an association between OPN levels and bone metastases in dogs [19]. The development of better therapeutics for PC bone metastasis requires in vivo animal models. Canine PC is uncommon in dogs and represents a unique translational model for cancer research [19].

Despite previous investigations in female dogs affected by MGTs [12,20], no association was found between OPN expression and bone metastasis or animal prognosis. However, in the human literature, the influence of OPN in determining a woman’s prognosis has been previously reported, and OPN is considered promising as a tissue and serum marker for mammary neoplasms [14]. In animals, previous studies have focused on the comparison of OPN expression and tumor histological characteristics; however, to our knowledge, no other studies have determined this protein as a bone metastasis marker in dogs.

Understanding the role of OPN in tumors prone to bone metastasis is important for the development of targeted therapies that can interfere with this process and improve the outcomes in animals with metastatic bone cancer. Owing to the lack of information regarding the association between OPN and canine patient prognosis and bone metastasis, this study aimed to investigate the association of OPN expression with prognosis and bone metastasis in dogs affected by MGTs and PC.

## 2. Materials and Methods

### 2.1. Study Design

This was a retrospective, nonrandomized study involving female and male dogs with MGTs and PC. Fifty animals (30 females and 20 males) were used in this study. Therefore, 30 primary MGTs from 30 female dogs were included in the present study. Among the female dogs, ten had MGTs with no metastasis at diagnosis, ten had MGTs with metastasis unrelated to the bones, and ten had bone metastasis. We only investigated OPN expression in primary tumors. Twenty male dogs with PC were selected: ten dogs with no evidence of metastasis at the time of diagnosis and ten dogs with bone metastasis at diagnosis. Tissue samples from primary tumors (*n* = 20) were used. Power analysis was performed using the G-power computer program (G-power^®^, Brunsbuttel, Germany). The following criteria were used to calculate the sample size: type II error rate (α) = 0.05 and type II error rate (β) = 0.2, with a statistical power of 80%. Thus, we determined a minimum of eight cases per group.

### 2.2. Inclusion Criteria

Only patients who presented with complete clinical information were selected, including epidemiological and clinical data, such as breed, age, treatment applied, and clinical follow-up. The patients underwent clinical staging and treatment according to the previous veterinary literature. After case selection, paraffin blocks were retrieved from the file, and only patients with sufficient tissue samples in the paraffin block for immunohistochemical analysis were included. For patients without bone metastasis, only radiographic examination was performed to exclude the presence of metastasis, or necropsy was performed with macroscopic assessment of skeletal bones demonstrating no evidence of bone metastasis.

For patients affected by MGTs, surgery was the only therapeutic option (even at metastatic sites when possible). For patients with PC, only those who were treated palliatively were included. Therefore, patients subjected to any specific antitumor treatment (chemotherapy or radiation therapy) were not enrolled in this study to avoid any influence on the survival analysis. 

### 2.3. Tumor Classification

Morphological diagnosis of canine MGTs was performed according to Zappulli et al. [21], and tumor grading was performed according to Peña et al. [22]. For canine PC, tumor classification was performed according to Palmieri et al. [23].

### 2.4. Immunohistochemistry Technique

To perform the immunohistochemistry, the polymer system (horseradish peroxidase (HRP)) and 3,3′-diaminobenzidine (DAB) as chromogen was applied [24]. The paraffin blocks were cut in a microtome to a thickness of 4 µm and extended on polymerized slides suitable for immunohistochemistry (StarFrost, Knittel, Braunschweig, Germany). The slides were stored for 24 h at a temperature of 55 °C drying oven and were transferred to vertical glass vats for the deparaffinization process. Antigen recovery was performed in a pressure cooker (Pascal, Dako Cytomation, Carpinteria, CA, USA) for approximately 1 h (with pH 6.0 citrate solution), and endogenous peroxidase blocking was performed with 8% hydrogen peroxide diluted in methyl alcohol. Nonspecific proteins were blocked with commercial Protein Block reagent (Dako Cytomation, Carpinteria, CA, USA) for 20 min. The slides were incubated with mouse monoclonal anti-OPN antibody clone LFMb-14 (Novocastra Laboratories, Newcastle upon Tyne, UK) at a dilution of 1:50. The Envision polymer system (Pascal, Dako Cytomation, Carpinteria, CA, USA) was used as the secondary antibody, and the reaction was visualized using DAB. Samples of normal decalcified bone tissue from patients undergoing necropsy were used as positive controls. Mouse immunoglobulin was used as the negative control at the same concentration as the primary antibody.

### 2.5. Data Analysis and Statistics 

Data were generated using semi-quantitative analyses. Immunohistochemistry was performed as previously described [25]. Briefly, distribution was categorized into scores of 0 (none), 1 (1–10%), 2 (11–33%), 3 (34–66%), and 4 (≥67%). The Chi-square or Fisher test was applied to evaluate the association of the IHC scores with clinical and pathological factors. Moreover, a Kaplan–Meier curve was generated to investigate the association between OPN expression patterns and survival. Owing to the small number of subjects, patients were grouped according to OPN expression for statistical purposes. Data were considered significant at *p* ≤ 0.05. To investigate whether OPN expression was truly associated with different clinicopathological findings, a multivariate matrix of correlations was constructed. Then, for data interpretation, the correlation coefficients (r) were interpreted based on the following intervals: weak (0–0.29), low (0.3–0.49), moderate (0.5–0.69), strong (0.7–0.89), or very strong (0.9–1.0), as well as whether they are positive or negative [26]. Statistical analyses were performed using GraphPad Prism (version 8.0; GraphPad Software Inc., La Jolla, CA, USA).

## 3. Results

The complete clinical data of the female patients enrolled in this study are shown in Table 1. The mean survival time for female dogs affected by MGTs with bone metastasis was 133.5 (±209.5) days, and for the female dogs with other metastasis, the mean survival time was 519 (±305.9) days, whereas the mean was 901.5 (±494.4) days for female dogs with no metastatic disease. For dogs with PC with bone metastasis, the mean survival time was 281.3 (±209.2) days. In contrast, patients with no metastatic disease had a mean survival of 2320.9 (±195.4) days. The complete clinical information for male dogs affected by PC is shown in Table 2.

Immunohistochemical analysis revealed diffuse cytoplasmic staining in 36.6% (11/30) of MGT samples. Higher OPN expression was detected in MGTs with bone metastasis (Figure 1A) than in the non-metastatic tumor group (Figure 1B) (*p* = 0.01). There was no statistically significant difference between the tumor group with other metastases and the non-metastatic tumor group (*p* = 0.786). Among the prostatic carcinomas without metastasis, 10% (1/10) showed positive OPN expression, whereas among the prostatic carcinomas with bone metastasis, 40% (4/10) showed positive OPN staining. Higher OPN expression was detected in samples from animals with primary tumors that presented with bone metastasis (Figure 1C) than in primary tumor samples without bone involvement (Figure 1D) (*p* = 0.0321). The contingency table for all data studied is available in the Appendix A.

To assess the potential role of OPN in the prognosis of canine tumors, we evaluated the association between OPN expression, disease-free survival, and overall survival. In mammary carcinoma samples, reduced disease-free time was not associated with higher OPN expression (*p* = 0.2010) (Figure 2A). However, regarding overall survival, we detected a significantly lower survival rate in patients with higher OPN expression (Figure 2B) (*p* = 0.0171). Prostatectomy was not performed in any animal patient with prostatic carcinoma. Therefore, we could not assess the DFS in any patient. Regarding overall survival, there was a strong trend toward lower survival in animals with positive OPN expression; however, this trend was not statistically significant (*p* = 0.0779) (Figure 2C).

In the multivariate analysis, we identified a strong correlation between patient age and histological subtype, with older patients presenting with more aggressive tumors (r = 0.74) (Figure 3). Interestingly, the patient age showed a moderate positive correlation with OPN expression. Therefore, older patients had a higher chance of developing tumors with aggressive histological subtypes (r = 0.57). Age also showed a strong negative correlation with the disease-free interval (r = −0.61) and overall survival (r = −0.64). OPN expression was negatively correlated with overall survival (r = −0.43), showed a weak positive correlation with histological subtype (r = 0.35), and was not correlated with tumor grade (r = 0). Therefore, the association between OPN expression and patient survival was independent of the clinicopathological features. We performed a multiple linear regression with each variable individually as a dependent variable and the others as independent variables. The results were similar to our multiple matrix correlation analysis (Figure 3); the complete logistic regression is shown in Appendix A.

## 4. Discussion

This study focused on the association between OPN expression and the bone metastasis phenotype. However, research with this focus is limited by the necessity of obtaining tissues from the primary tumor and the respective bone metastases. Because we aimed to apply restrictive exclusion and inclusion criteria, only a small number of patients were enrolled. Therefore, we applied a power analysis to ensure that our set of patients would be statistically significant. In addition, as our patients should have undergone necropsy or for any previous reason to undergo X-ray screening to exclude bone metastasis, few patients met the study criteria, even when we considered our control group (patients with the respective tumor but without metastasis). 

Metastasis is mediated through interactions between the host tissue microenvironment components and invading cell tumor-secreted molecules, which corroborate the stromal remodeling and invasion processes [27]. Consequently, skeletal-related events can lead to devastating complications, such as bone pain, pathological fractures, hypercalcemia, and spinal cord compression [28].

Bone metastasis is very uncommon in canine MGTs and most frequently occurs in canine PC. Although bone metastasis is common in PC, it is considered rare, and it is very difficult to achieve a large number of tumors with bone metastasis using only PC cases. Therefore, we grouped prostate cancer and MGTs together. In veterinary medicine, there is no widely used method to precisely identify bone metastases, and bone scintigraphy appears to be the best option [29]. Therefore, it was difficult to use our inclusion criteria for patients without bone metastases to ensure that they had no microscopic lesions. For cases with no bone metastasis, we used general clinical signs, the absence of any sign related to the osteoarticular system, and X-rays of limbs and column or necropsy (when available). Although this method may not be the gold standard for patient selection, other diagnostic methods for accurately identifying bone metastasis are unknown. According to previous veterinary literature, whole computed tomography is not suitable for identifying bone lesions [29]. According to these authors, bone scintigraphy is the better method for examining bone lesions. However, we were unable to perform this examination.

Once again, it is important to highlight that the bone is not a common site for MGT metastasis and is very unlikely in patients with metastatic lesions. For cases of PC, X-rays of the column, limbs, and pelvis were performed for all patients at the time of diagnosis as well as in association with clinical signs and patient history. In PC, there is a high probability of bone metastasis, although radiography or necropsy may not detect metastasis. 

OPN contains 300 amino acids, some modified with O-N-linked oligosaccharides, and it is a highly phosphorylated glycophosphoprotein rich in aspartic acid [30]. OPN has pleiotropic effects, including cell adhesion, migration, proliferation, survival, differentiation, and activation in various cell types such as epithelial and endothelial cells, fibroblasts, and immune cells. The interaction of OPN with a variety of cell surface receptors can result in the activation of various signal transduction pathways, promoting changes in gene expression that influence cell behavior, such as invasion and metastasis, leading to enhanced proliferation and survival rates. Increased proliferation and tumor cell survival rates may also be related to drug resistance in different types of cancer. In addition, OPN is responsible for interactions with host defense systems, leading to enhanced survival of tumor cells, thus playing an important role in the resistance to tumor killing by the immune system [31]. 

Breast and PC have been linked to high expression levels of the bone matrix protein OPN, and its expression has been negatively correlated with patient survival in retrospective studies. For example, in an immunocytochemical study of primary human breast cancers, OPN was shown to promote breast cancer dissemination [32]. These findings corroborate the results of the present study, which showed higher OPN expression rates in MGTs with bone metastasis, as described by Carlifante et al. [33]. In addition, a lower mean survival time to bone metastasis has been reported in patients with bone metastasis than in those without or with other metastases [33]. Interestingly, higher OPN expression in samples from primary tumors with bone metastasis could also be related to poor prognosis, as shown in a study by Zduniak et al. [6], who investigated the expression of OPN in tissue samples from patients with clear cell renal cell carcinoma (ccRCC) and its potential role as a prognostic marker for the disease, and they provided important insights into the role of OPN in ccRCC, suggesting that OPN expression may be a useful prognostic marker for the disease.

Canine PC typically occurs spontaneously in middle-aged or older adults and is usually androgen independent. It often metastasizes to the lungs, regional lymph nodes, and very frequently to the bones [34]. Substantial efforts have been made to identify and isolate genetic factors involved in the malignant progression of PC. In previous studies, increased OPN expression has been implicated in the malignant transformation of prostate epithelial cells, tumor progression [16] and patient survival [32]. In the present study, we reported higher OPN expression in samples from primary tumors with bone metastasis than in those lacking metastases, supporting the role of this protein in cancer cell progression, which is in agreement with human studies [16,35]. Our findings are similar to those reported by Kim et al. [36] and Bramwell et al. [15], who showed that human patients with metastatic cancer have higher circulating levels of OPN and tumors that are more likely to metastasize.

The lower overall survival in patients with PC was directly correlated to positive OPN expression as demonstrated by Caruzo et al. [37], which investigated the expression of several biomarkers, including OPN, in PC tissue samples from 161 patients and their correlation with disease progression and overall survival, showing that higher OPN expression in PC was linked to poor patient prognosis. This could be explained by the paracrine activity mediated by OPN through the induction of pro-inflammatory molecules, which may be responsible for a reactive stromal pattern leading to tumor progression and tissue remodeling.

Based on these findings, we conclude that OPN is a promising prognostic target for mammary cancer. In our study, both MGTs and PC demonstrated 32% (16/50) positive immunohistochemical staining for OPN, and this expression is variable according to the literature due to the bone metastasis subtype because the matrix proteins have different levels of osteotropic phenotype [33]. Further studies are required to determine the association between OPN expression and prognosis.

Several studies have shown that OPN plays crucial roles in tumor progression, angiogenesis, and metastasis in these cancers. For instance, Tan et al. [38] demonstrated that high OPN expression in breast cancer cells promotes tumor growth and invasiveness. In PC, OPN plays a critical role in the development of castration-resistant diseases that are particularly challenging to treat [35]. However, more research is needed to fully understand the molecular mechanisms underlying the role of OPN in breast and PC and to develop effective OPN-targeted therapies. Overall, the evidence suggests that OPN is a promising prognostic target for MGTs and PC, and further research in this area is warranted.

## 5. Conclusions

Owing to the lower overall survival related to OPN-positive cases, it is possible to hypothesize a link between the presence of OPN expression and a poor prognosis. From these findings, it can be concluded that OPN may be a promising prognostic target for MGTs and PC owing to its role in enhancing cell invasion and metastasis.

## Figures and Tables

**Figure 1 animals-13-03211-f001:**
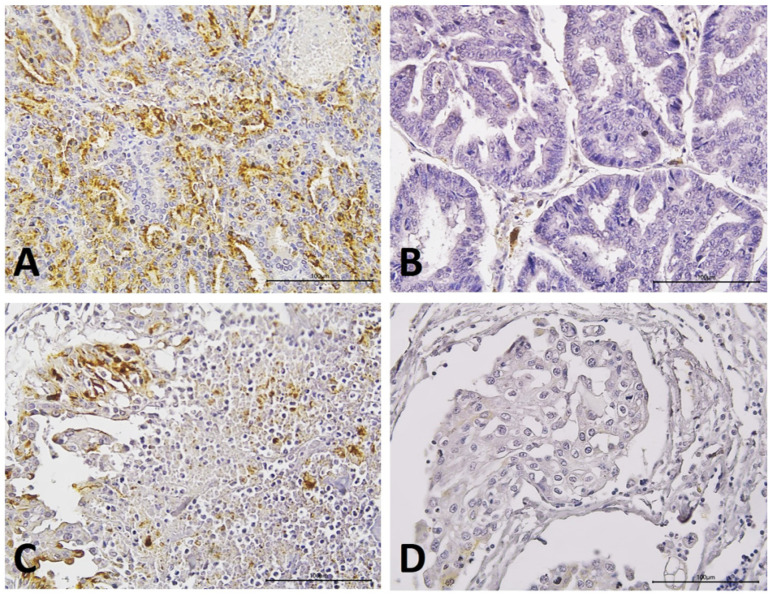
Immunohistochemical staining of osteopontin (OPN) in mammary gland and prostatic carcinoma samples from dogs. (**A**): Cytoplasmic and membranous staining of OPN in a sample of mammary gland tumor (MGT) that has metastasized to bone. (**B**): Absence of OPN staining in a sample of mammary carcinoma with no bone metastasis. (**C**): Positive OPN staining in a prostate carcinoma with bone metastasis. (**D**). Absence of OPN staining in a sample of prostate carcinoma without metastasis.

**Figure 2 animals-13-03211-f002:**
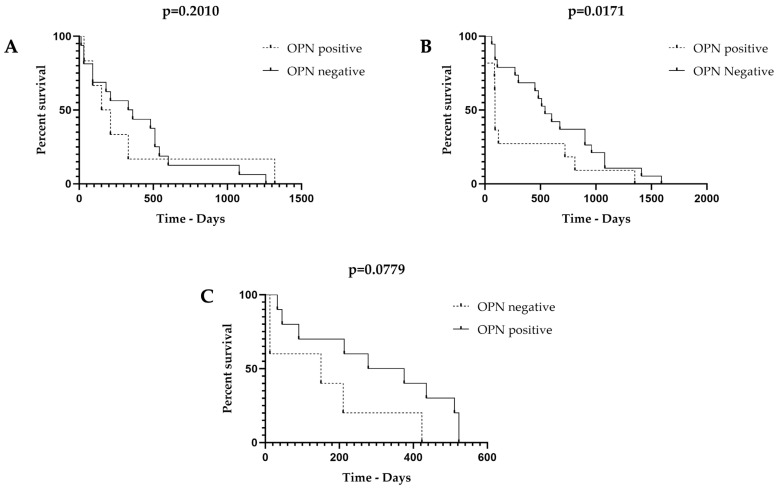
A: Patients disease-free intervals based on osteopontin (OPN) expression. (**A**): No association was found between OPN expression and disease-free interval for patients with mammary gland tumors (MGTs). (**B**): Reduced survival time was observed in patients with positive OPN expression relative to patients lacking OPN expression, and this difference was statistically significant in patients with MGTs. (**C**): There was no statistical difference between the overall survival in OPN positive or negative expression in patients with prostate cancer (PC).

**Figure 3 animals-13-03211-f003:**
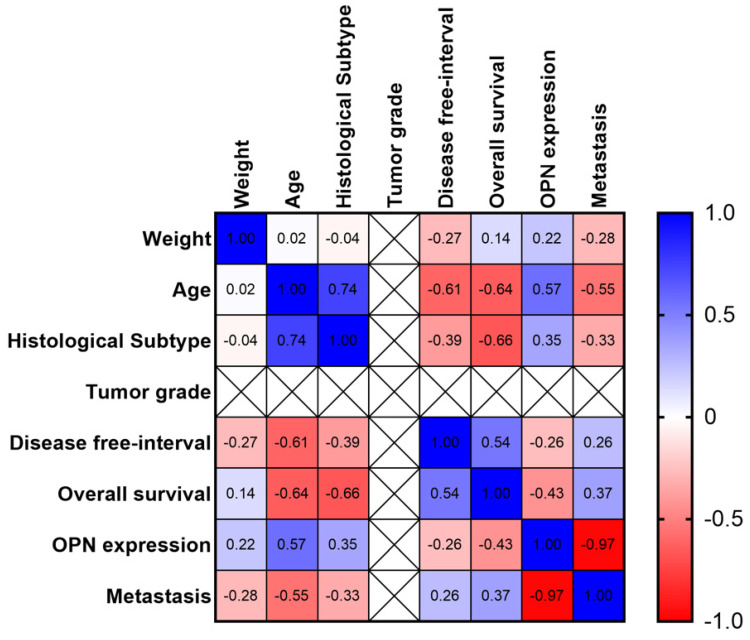
Multivariate correlation matrix of the clinic–pathological and osteopontin (OPN) expression in canine tumors. Blue indicates a positive correlation and red indicates a negative correlation. The color intensity is associated with the strength of the correlation (higher intensity indicates stronger correlation).

**Table 1 animals-13-03211-t001:** Clinic-pathological information of female dogs affected by mammary gland tumors (MGTs).

	Breed	Weight (kg)	Age (years)	Tumor Subtype	Grade	Disease Free Interval (d)	Overall Survival (d)	OPN Expression
Female dogs with bone metastasis	Dogo Argentino	N/A	12	Solid carcinoma	III	0	1	3
	Rottweiler	27	N/A	Anaplastic carcinoma	III	0	1	3
	German Shepherd	34	11	Invasive micropapillary carcinoma	II	0	720	2
	Pinscher	3.3	11	Anaplastic carcinoma	III	12	60	0
	Poodle	6	12	Solid carcinoma	III	90	90	3
	Brazilian Bullmastiff	25.4	9	Solid carcinoma	II	0	90	4
	Pitbull	30.5	6	Mixed carcinoma	II	150	90	3
	Poodle	4	12	Anaplastic carcinoma	III	0	111	0
	Poodle	6	9	Carcinosarcoma	II	32	87	4
	Lhasa Apso	4	8	Solid carcinoma	III	0	85	3
Female dogs with other metastases (excluding bone)	Beagle	10.5	2	Solid carcinoma	I	210	480	0
	Mixed breed	N/A	N/A	Complex carcinoma	III	480	510	0
	Teckel-Dachshund	8	3	Carcinosarcoma	II	30	90	0
	Teckel-Dachshund	10	2	Tubulopapillary carcinoma	III	30	270	0
	Poodle	6.5	2	Mixed carcinoma	I	0	900	0
	Pitbull	35.4	2	Mixed carcinoma Tubulopapillary carcinoma	I	1	960	0
	Akita	19	3	Tubular carcinoma	II	11	120	2
	Beagle	16.4	2	Mixed carcinoma	I	213	810	2
	Teckel-Dachshund	6.1	2	Mixed carcinoma	II	517	600	0
	Mixed breed	5.8	2	Tubulopapillary carcinoma	II	335	450	0
Female dogs with no metastasis	Pinscher	3.6	8	Solid carcinoma	II	609	675	0
	American Cocker	20	1	Complex carcinoma	I	1096	1080	0
	Mixed breed	8.1	2	Mixed carcinoma	I	183	1590	0
	Mixed breed	1.4	1	Mixed carcinoma	I	517	900	0
	Poodle	4	2	Tubulopapillary carcinoma	I	1340	1350	3
	Maltese	2.5	1	Tubulopapillary carcinoma	I	548	540	0
	Poodle	10.5	2	Tubulopapillary carcinoma	I	0	300	0
	Poodle	5.4	1	Mixed carcinoma	I	91	90	0
	Mixed breed	7	1	Tubulopapillary carcinoma	I	1279	1410	0
	Mixed breed	15.5	2	Complex carcinoma	I	91	1080	0

**Table 2 animals-13-03211-t002:** Clinic-pathological information from dogs affected by prostate cancer (PC). Not applicable (N/A).

	Breed	Age (Years)	Metastatic Site	Histological Pattern	Gleason Scale	Overall Survival (day)	OPN Expression
Canine patient with PC with no bone metastasis	German Shepherd	10	No site	Acinar small and cribriform	8	N/A	0
	Teckel	11	No site	Cribriform	10	N/A	0
	Poodle	8	No site	Acinar small	6	45	0
	American Cocker	10	No site	Acinar small	10	32	0
	Mixed breed	9	No site	Acinar small	6	523	0
	Poodle	10	No site	Solid	10	213	0
	Mixed breed	13	No site	Cribriform	10	N/A	0
	Poodle	14	No site	Acinar small	6	435	0
	Mixed breed	10	No site	Acinar small	6	511	0
	German Shepherd	13	No site	Solid	10	210	2
Canine patient with PC with bone metastasis	Mixed breed	9	Bone	Cribriform	10	150	3
	Boxer	11	Bone	Acinar small	6	523	0
	American Pitbull	10	Bone and Lung	Acinar small	6	N/A	0
	Mixed breed	13	Bone, Intestine, Liver	Acinar small and cribriform	8	12	3
	Mixed breed	15	Bone and Lung	Cribriform	10	423	2
	Boxer	12	Bone and Lung	Cribriform	10	278	0
	Boxer	14	Lung, bone and Intestine	Cribriform	10	90	0
	Terrier	10	Bone and Lung	Acinar small	6	N/A	0
	German Shepherd	12	Bone and Lung	Solid	10	375	0
	Mixed breed	13	Bone, Intestine and Lung	Acinar small and cribriform	8	12	3

## Data Availability

Data sharing not applicable.

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
