# Peer review of "Is Osteopontin a Good Marker for Bone Metastasis in Canine Mammary Gland Tumor and Prostate Cancer?"

_animals, 2023, doi:10.3390/ani13203211_

Round 1

Reviewer 1 Report

The manuscript presents the results of a study aimed to describe the Osteopontin (OPN) immunohistochemical expression in canine mammary tumors (CMT) and prostate cancer (PC) with or without bone metastasis. FFPE primary and metastatic neoplastic tissue samples from 50 dogs were included in the study. The introduction is focused on the multifarious mechanisms in which OPN is involved and is informative, still, the bibliography could be updated and should be checked to be opportunely cited. The metastatic process in CMT and PC represents undoubtedly a pivotal research topic in human and canine tumor growth and further studies in this field are needed. The methodology is straight; however, the theoretical background should be implemented, especially from a veterinary perspective. English language and style are fine, but a revision from a native English speaker is recommended. In conclusion, I believe that this manuscript could be accepted for publication only after some major revisions.

Major revisions:

Introduction. This section is generally informative, however, it’s not always clear when the authors refer to human rather than veterinary malignancies. Since the authors focused on the mechanisms played by the OPN molecule in human malignancies, they should add more recent and updated references on this topic and explain better when they are discussing human, animal, or in vitro neoplasia. The text is a bit redundant and a native English speaker revision as long as some thinning should improve the value of the manuscript. Moreover, some references seem to be displaced (#3, #12, #15-17), so a throughout bibliography check is suggested.

Page 2 line 49. Concerning the sentence “After the lungs and liver, the skeleton is the most common site…”, the authors should provide a reference for this claim. Moreover, reference #4 is lacking in the text and is incorrectly positioned in the bibliography.

Page 2 lines 56-57. This sentence on veterinary medicine is isolated from the rest of the veterinary medicine background. I suggest moving it to the beginning of lines 88-93.

Page 2 line 74. The reference #12 is a morphofunctional study of three mammary glands from Raccoons. Why do the authors include it?

Page 2 line 96. The references #15 and 16 are not about PC and OPN expression.

Page 3 line 108. The reference #17 is not about canine PC and more references on this topic should be added.

Page 3 lines 139-141. What is the effect size considered by the authors in the power analysis? How was it estimated?

Page 5 lines 199 and 201 (Table 1; Table 2). The authors should add at least two columns regarding the therapy to which each animal was subjected, and the OPN expression resulting by IHC. If all the animals received the same therapeutic iter should made explicit in the text. In addition, bone metastases localization (lumbar, ribs, etc..) should be added, if available.

Alternatively, the OPN expression in the different samples could be added to a new table, together with the standard deviation.

Discussion. In this section, as for the introduction, some parts seem to be redundant (es lines 255-262), while more focus should be put on veterinary or laboratory animals literature. Some of these papers are already cited (es #13 and #18) and should be stressed more, while other papers on IHC or molecular investigation on OPN in non-human species should be added.

Moreover, given the amount of data taken from BC and human PC, the biological differences between them and their canine counterparts should be discussed.

Minor revisions:

Page 1 lines 25-26. If there is no study associating OPN with metastasis, the part “…of canine mammary gland tumors (CMT) and prostate cancer (PC)” could be deleted.

Page 1 lines 27-28. “OPN expression was immunohistochemically evaluated in metastatic and non-metastatic FFPE tissue samples from 50 subjects.”

Page 2 line 45. “…and is often associated with a poor prognosis”

Page 2 lines 45-48. At least one reference should be added.

Page 2 lines 49-50. Are the authors sure of this claim? Could a reference to this claim be included?

Page 2 line 55. At least one reference should be added.

Page 2 line 57. When referring to human or in vitro literature, why do the authors state that there are few studies?

Page 2 line 88. The part “However, regarding the OPN expression..” could be deleted.

Page 2 line 89. The part “Addressing this problem,..” could be deleted.

Page 2 line 91. I suggest rephrasing the sentence. I don't think “frequent” is the best term to describe bone metastases in dogs with CMT or PC. However, I agree with the author when they say that among bone metastasis, CMT and PC are some of the most common primary neoplasia at work.

Page 3 line 100. At least one reference should be added.

Page 3 lines 115-116. This sentence is a repetition of what was previously said in lines 103-105, and lines 88-89.

Page 3 lines 117-118. The authors should clarify better the meaning of this sentence.

Page 3 line 127. “..including female and male dogs affected…”

Page 3 line 128. “Fifty”

Page 3 line 133. Shouldn't there be 30 primary tumors (10 non-metastatic, 10 with non-bone metastasis, and 10 with bone metastasis), and 20 metastasis?

Page 4 line 183. “Kaplan-Meier”

Page 7 line 221. Why do the authors write about human tumors’ prognosis?

Page 8 line 233. In Figures 2B and 2C should be clarified if the curves refer to CMT or PC population. The caption should be checked, too.

Page 9 lines 292-299. The authors should state explicitly that the papers to which they are referring (#27,29,16,30) are on human PC.

Page 9 line 306. Given the scant data in the veterinary literature and the reduced number of animals investigated, the authors should use more hedging language when discussing these preliminary results. In the conclusions, softening claims should be appropriate, too.

Page 9 lines 317-319. Since the authors performed only IHC, which does not provide any direct information on molecular mechanisms, if not phenotypic, this limitation could be included in the discussion, too.

Page 10 lines 322-324. The authors should rephrase the sentence.

Extensive editing of English language required

Author Response

Reviewer 1.

The manuscript presents the results of a study aimed to describe the Osteopontin (OPN) immunohistochemical expression in canine mammary tumors (CMT) and prostate cancer (PC) with or without bone metastasis. FFPE primary and metastatic neoplastic tissue samples from 50 dogs were included in the study. The introduction is focused on the multifarious mechanisms in which OPN is involved and is informative, still, the bibliography could be updated and should be checked to be opportunely cited. The metastatic process in CMT and PC represents undoubtedly a pivotal research topic in human and canine tumor growth and further studies in this field are needed. The methodology is straight; however, the theoretical background should be implemented, especially from a veterinary perspective. English language and style are fine, but a revision from a native English speaker is recommended. In conclusion, I believe that this manuscript could be accepted for publication only after some major revisions.

Answer: Thank you so much for the positive criticisms and comment regarding our manuscript. The reviewer’s vision on our manuscript was very interesting and pivotal for increasing manuscript quality. We also submitted our manuscript for English review by Editage. Thank you so much. Please, see below the specific comments regarding the reviewer’s issues.

Major revisions:

Introduction. This section is generally informative, however, it’s not always clear when the authors refer to human rather than veterinary malignancies. Since the authors focused on the mechanisms played by the OPN molecule in human malignancies, they should add more recent and updated references on this topic and explain better when they are discussing human, animal, or in vitro neoplasia. The text is a bit redundant and a native English speaker revision as long as some thinning should improve the value of the manuscript. Moreover, some references seem to be displaced (#3, #12, #15-17), so a throughout bibliography check is suggested.

Answer: Dear reviewer, we have included two new recent references (both from 2023) an we  have adjust the introduction regarding point out information of human and canine cancers as suggested. 

Page 2 line 49. Concerning the sentence “After the lungs and liver, the skeleton is the most common site…”, the authors should provide a reference for this claim. Moreover, reference #4 is lacking in the text and is incorrectly positioned in the bibliography.

Answer: Sorry for this mistake, we have provided a reference and have modified in the reference list the references 3 and 4.

Page 2 lines 56-57. This sentence on veterinary medicine is isolated from the rest of the veterinary medicine background. I suggest moving it to the beginning of lines 88-93.

Answer: thank you so much for the suggestion. We have modified as suggested.

Page 2 line 74. The reference #12 is a morphofunctional study of three mammary glands from Raccoons. Why do the authors include it?

Answer: we are sorry for this confusion. This was made on the first review provided by MDPI office, asking for veterinary studies in the manuscript and we misinterpreted some parts. We have adjusted and included an appropriated reference.

Page 2 line 96. The references #15 and 16 are not about PC and OPN expression.

Answer: Sorry for this mistake, we have adjusted as suggested.

Page 3 line 108. The reference #17 is not about canine PC and more references on this topic should be added.

Answer: We are sorry for the confusion. In a first review report, MDPI office requested to add more references and we did a confusion in the number. However, we have adjusted, thank you for your comments.

Page 3 lines 139-141. What is the effect size considered by the authors in the power analysis? How was it estimated?

Answer: we have evaluated the effect size by the ratio of “population” standard deviations. Our population was based on the archive of Veterinary Pathology service. We have calculated around 600 cases from the past 10 years with only 10 out of 609 cases showing bone metastasis.

Page 5 lines 199 and 201 (Table 1; Table 2). The authors should add at least two columns regarding the therapy to which each animal was subjected, and the OPN expression resulting by IHC. If all the animals received the same therapeutic iter should made explicit in the text. In addition, bone metastases localization (lumbar, ribs, etc..) should be added, if available.

Answer: we have added columns to the table as suggested. For mammary gland tumors. Only surgery was performed with no adjuvant treatment.

Alternatively, the OPN expression in the different samples could be added to a new table, together with the standard deviation.

Answer: we have added as suggested.

Discussion. In this section, as for the introduction, some parts seem to be redundant (es lines 255-262), while more focus should be put on veterinary or laboratory animals literature. Some of these papers are already cited (es #13 and #18) and should be stressed more, while other papers on IHC or molecular investigation on OPN in non-human species should be added.

Moreover, given the amount of data taken from BC and human PC, the biological differences between them and their canine counterparts should be discussed.

Answer: we have added the discussion as suggested.

Minor revisions:

Page 1 lines 25-26. If there is no study associating OPN with metastasis, the part “…of canine mammary gland tumors (CMT) and prostate cancer (PC)” could be deleted.

Answer: we have adjusted the phrase to be more clear.

Page 1 lines 27-28. “OPN expression was immunohistochemically evaluated in metastatic and non-metastatic FFPE tissue samples from 50 subjects.”

Answer : we have added as suggested.

Page 2 line 45. “…and is often associated with a poor prognosis”

 Answer: we have added as suggested.

Page 2 lines 45-48. At least one reference should be added.

Answer: we have added as suggested.

Page 2 lines 49-50. Are the authors sure of this claim? Could a reference to this claim be included?

 Answer: we deleted the phrase.

Page 2 line 55. At least one reference should be added.

 Answer: we have added as suggested.

Page 2 line 57. When referring to human or in vitro literature, why do the authors state that there are few studies?

 Answer: we have deleted this phrase.

Page 2 line 88. The part “However, regarding the OPN expression..” could be deleted.

 Answer: we have added as suggested.

Page 2 line 89. The part “Addressing this problem,..” could be deleted.

 Answer: we have added as suggested.

Page 2 line 91. I suggest rephrasing the sentence. I don't think “frequent” is the best term to describe bone metastases in dogs with CMT or PC. However, I agree with the author when they say that among bone metastasis, CMT and PC are some of the most common primary neoplasia at work.

 Answer: We modified the phrase.

Page 3 line 100. At least one reference should be added.

Answer: we added a reference.

Page 3 lines 115-116. This sentence is a repetition of what was previously said in lines 103-105, and lines 88-89.

Answer; We deleted the phrase.

Page 3 lines 117-118. The authors should clarify better the meaning of this sentence.

Answer: we have clarified as suggested.  

Page 3 line 127. “..including female and male dogs affected…”

 Answer: we have clarified as suggested.  

Page 3 line 128. “Fifty”

 Answer: we have clarified as suggested.  

Page 3 line 133. Shouldn't there be 30 primary tumors (10 non-metastatic, 10 with non-bone metastasis, and 10 with bone metastasis), and 20 metastasis?

Answer: Should be 50 dogs, being 30 affected by mammary gland tumors and 20 for prostate cancer. The 20 metastasis should not be here. We have excluded because we do not evaluated metastatic tissues.

Page 4 line 183. “Kaplan-Meier”

Answer: we have adjusted as suggested.

Page 7 line 221. Why do the authors write about human tumors’ prognosis?

 Answer: sorry for the mistake. Should be “canine”. We have adjusted.

Page 8 line 233. In Figures 2B and 2C should be clarified if the curves refer to CMT or PC population. The caption should be checked, too.

 Answer: We have adjusted.

Page 9 lines 292-299. The authors should state explicitly that the papers to which they are referring (#27,29,16,30) are on human PC.

 Answer: We have clarified that were human patients.

Page 9 line 306. Given the scant data in the veterinary literature and the reduced number of animals investigated, the authors should use more hedging language when discussing these preliminary results. In the conclusions, softening claims should be appropriate, too.

 Answer: we were sorry for the strong statement regarding OPN and canine cancers. We have modified all of them and suggested only as a potential prognostic marker.

Page 9 lines 317-319. Since the authors performed only IHC, which does not provide any direct information on molecular mechanisms, if not phenotypic, this limitation could be included in the discussion, too.

 Answer: thank you for the suggestion. We have included in the manuscript.

Page 10 lines 322-324. The authors should rephrase the sentence.

Answer: we have modified.

Reviewer 2 Report

I read the manuscript “Is Osteopontin a good marker for bone metastasis in canine mammary gland tumors and prostate cancer?”. The study deals with the studying of immunohistochemical expression of Osteopontin in such these kind of tumors in two different canine populations, one bearing mammary gland tumors and the other with prostate cancer. The work is not so original, since there are already other works in veterinary literature, even if the authors are aware of this and cited one or two of them throughout the text. Nevertheless, the issue is interesting, so further quality works on these prognostic markers is to be encouraged. The title is appropriated, the introduction supports the research study, however there are some critical flaws in the study design -or in the results presentation, it is not clear- and consequently discussions section also need to be reviewed. Below my main concerns.

First of all, in lines 133-136 the authors declare that 20 primary mammary gland tumors and their metastasis were enrolled in the study, however they do not clarify according to which criterion this choice was made and whether the twenty samples chosen are tumors without metastases or with metastases and, in this case, in which site. Not even the attached table shows this information, which is essential. Moreover, they state that they have analyzed the expression of the marker both in primary tumors and in metastases, but there is no trace of the results of the analyzes of the metastases in the results section, this whole part is missing, there isn’t a qualitative description of immunohistochemical results, nor a statistical analysis or pictures about metastasis samples. In addition, this is a work evaluating the expression of a prognostic marker, in my opinion is a little reckless to state that through this research we can conclude that Osteopontin may be a good therapeutic marker as they stated in LL 306-311 and LL 319-320. It can certainly be included among the markers to be investigated as a therapeutic target, but we cannot ascertain its usefulness as therapeutic target from such a pathological study. The authors should remodel this concept. 

Moderate editing of English language is required. 

Author Response

Reviewer 2.

Comment: I read the manuscript “Is Osteopontin a good marker for bone metastasis in canine mammary gland tumors and prostate cancer?”. The study deals with the studying of immunohistochemical expression of Osteopontin in such these kind of tumors in two different canine populations, one bearing mammary gland tumors and the other with prostate cancer. The work is not so original, since there are already other works in veterinary literature, even if the authors are aware of this and cited one or two of them throughout the text. Nevertheless, the issue is interesting, so further quality works on these prognostic markers is to be encouraged. The title is appropriated, the introduction supports the research study, however there are some critical flaws in the study design -or in the results presentation, it is not clear- and consequently discussions section also need to be reviewed. Below my main concerns.

Answer: Dear reviewer, thank you so much for you kind comments. We have modified our manuscript, including a stronger multifactorial statistical analysis that confirmed OPN expression being an independent prognostic factor for patient with canine mammary gland tumors. Thank you so much for your time and kind comment in our manuscript.

First of all, in lines 133-136 the authors declare that 20 primary mammary gland tumors and their metastasis were enrolled in the study, however they do not clarify according to which criterion this choice was made and whether the twenty samples chosen are tumors without metastases or with metastases and, in this case, in which site. Not even the attached table shows this information, which is essential. Moreover, they state that they have analyzed the expression of the marker both in primary tumors and in metastases, but there is no trace of the results of the analyzes of the metastases in the results section, this whole part is missing, there isn’t a qualitative description of immunohistochemical results, nor a statistical analysis or pictures about metastasis samples.

Answer: Dear reviewer, we are so sorry for providing a confusing presentation of the manuscript. We are really sorry. We have provided a more clear inclusion and exclusion criteria, for a better understanding of our selection. Regarding the metastasis, it was our first goal include the respective metastatic samples. However when we look at our archive, we do not find a sufficient number of paraffin blocks for analysis. We have adjusted and have excluded mentions related to the metastatic samples.

In addition, this is a work evaluating the expression of a prognostic marker, in my opinion is a little reckless to state that through this research we can conclude that Osteopontin may be a good therapeutic marker as they stated in LL 306-311 and LL 319-320. It can certainly be included among the markers to be investigated as a therapeutic target, but we cannot ascertain its usefulness as therapeutic target from such a pathological study. The authors should remodel this concept. 

Answer: the reviewer is completely right. We are sorry for providing a so strong statement in our manuscript. We have modified.  

Reviewer 3 Report

The manuscript reviewed summarizes an evaluation of banked canine cancer tissues for presence of osteopontin as a prognostic indicator in cases of mammary cancer and prostate cancer.  The experimental design is straightforward and sufficient.  The manuscript would benefit from revisions to address the following points:

1. Under 2.1, "Study Design," the second line states, "Fifth animals were enrolled..."  Presumably, you mean "fifty."

2. Under section 2.1, Study Design, it appears that 30 females were enrolled in the MGT evaluation, with 10 each in groups with no metastasis or metastasis to sites other than bone or with bone metastasis.  However, Line 133 states that 20 primary MGT tissues were evaluated.  I find this confusing - shouldn't this be a total of 30 primary tumors and 20 metastatic tissues?

3. It would be helpful to present the data for percent of samples staining positive for each treatment group and tumor type in the form of a table.

4. Under Results, Line 221, you state "To assess the potential role of OPN in the prognosis of human tumors, we evaluated..."   Were human tissue samples also evaluated or do you mean "canine" tumors?

5. For Figure 2, the legend should state which tumor type is represented (prostate cancer in this case).  Also, it would be helpful to have a Kaplan-Meier for the MGT data.

6. In the Conclusion, the authors suggest that the data show an association between OPN and diagnosed bony metastasis as a prognostic indicator; however, the more meaningful value of OPN as a prognostic marker would be if presence of OPN was found in primary tumors prior to diagnosis of bony metastasis. That is, if tumors become OPN positive once metastasis has occurred, can that finding be used to monitor animals for bony metastasis.  The authors should consider addressing this aspect of the data somewhere in the Discussion.

7. In the Discussion, line 306/307, the authors state that "From the findings presented, we conclude that OPN may be a promising anticancer therapy target..."  In contrast, the title of the manuscript and the Introduction emphasize, more appropriately, that the purpose of the study is to establish the potential utility of OPN as a prognostic biomarker.  I understand the enthusiasm of the authors for the possibility of OPN as a therapeutic target, however, it may be a bit strong to state this as a conclusion of the study as presented.

There are some minor improvements needed to the quality of the English used.  For example, "Veterinary Medicine" does not need to be capitalized.

Author Response

Reviewer 3:

The manuscript reviewed summarizes an evaluation of banked canine cancer tissues for presence of osteopontin as a prognostic indicator in cases of mammary cancer and prostate cancer.  The experimental design is straightforward and sufficient.  The manuscript would benefit from revisions to address the following points:

Answer: Dear reviewer, thank you so much for your comments and positive criticisms for our manuscript improvement. We have included all recommendations and we also performed an extensive English review using the international English service Editage. Therefore, we hope this new version it will be more suitable for publication.

Under 2.1, "Study Design," the second line states, "Fifth animals were enrolled..."  Presumably, you mean "fifty."

Answer: we are sorry for this mistake. We have adjusted.

Under section 2.1, Study Design, it appears that 30 females were enrolled in the MGT evaluation, with 10 each in groups with no metastasis or metastasis to sites other than bone or with bone metastasis.  However, Line 133 states that 20 primary MGT tissues were evaluated.  I find this confusing - shouldn't this be a total of 30 primary tumors and 20 metastatic tissues?

Answer: we are sorry for this confusion. Unfortunately, we tried to have access to the metastatic sites. However, we were not able to find most of them. Then, we excluded from the research work. We have adjusted.

It would be helpful to present the data for percent of samples staining positive for each treatment group and tumor type in the form of a table.

Answer: Thank you for the suggestion, we have provided a table with all patient information.

Under Results, Line 221, you state "To assess the potential role of OPN in the prognosis of human tumors, we evaluated..."   Were human tissue samples also evaluated or do you mean "canine" tumors?

Answer: we are sorry for this typo. Yes, we were meaning canine tumors.

For Figure 2, the legend should state which tumor type is represented (prostate cancer in this case).  Also, it would be helpful to have a Kaplan-Meier for the MGT data.

Answer: We are sorry for this confusion. The figures A and B are from MGT and the figure C for prostate cancer. We have adjusted the figure caption.

In the Conclusion, the authors suggest that the data show an association between OPN and diagnosed bony metastasis as a prognostic indicator; however, the more meaningful value of OPN as a prognostic marker would be if presence of OPN was found in primary tumors prior to diagnosis of bony metastasis. That is, if tumors become OPN positive once metastasis has occurred, can that finding be used to monitor animals for bony metastasis.  The authors should consider addressing this aspect of the data somewhere in the Discussion.

Answer: Thank you for the suggestion, we have included this information in discussion section.

In the Discussion, line 306/307, the authors state that "From the findings presented, we conclude that OPN may be a promising anticancer therapy target..."  In contrast, the title of the manuscript and the Introduction emphasize, more appropriately, that the purpose of the study is to establish the potential utility of OPN as a prognostic biomarker.  I understand the enthusiasm of the authors for the possibility of OPN as a therapeutic target, however, it may be a bit strong to state this as a conclusion of the study as presen

Answer: The reviewer is completely right. It was a really strong statement and we have excluded this statements from alongside the manuscript. Thank you for your suggestion.

Reviewer 4 Report

Analyzing osteopontin in tumor entities with frequent bone metastasis in dogs is an interesting approach and the results are indeed promising. However, some points need to be improved.

Abstract:

Canine mammary gland tumor is abbreviated CMT here, whereas in the text MGT is used.

Here, as well as in the whole manuscript, information about the expression in primary tumors and metastasis is lacking.

Introduction:

Overall, the introduction section is quite long for one analyzed marker and in some parts redundant.

Line 41: Consider the wording: The metastatic cascade is not literally a physiologically coordinated process; it is rather a selection or evolvement of tumor cells, that randomly feature the mechanisms needed for all the mentioned steps of metastasis. Another example is in line 50, “radiological symptoms”. Symptoms are clinical, however, metastasis can be detected radiologically.

Lines 100-108: The passage about canine prostate cancer should be supported by some references. Furthermore, there is an issue with reference No. 17, Line 108, this reference does not state, that PC is common in dogs. This statement itself is a bit difficult, since there is still a lot research to do regarding prevalence and early stages. It would fit much better to MGT, which definitely is a common problem in veterinary medicine. I would also suggest a short sentence introducing canine MGT.

Material and Methods:

I have some questions regarding the study design:

-          How was the sampling performed, were those biopsies, or samples obtained from mammectomies? Were the metastasis samples taken at the same timepoint as the primary tumors? In which time period did the sampling take place and in which veterinary hospital(s)? How was the follow-up information collected; were the owners contacted? Line 128: Fifth = fifty?, Line 139: type 9 error = type I error?

Inclusion criteria:

-          I would exclude the age from the complete clinical information, since the age is not available for some of the patients in Table 1 and 2.

-          If there is complete information about the treatment, why was it not included in the results? The choice of treatment could have a major influence on clinical outcome and survival times.

-          How was the histopathological classification performed, according to which guidelines/publications?

Data analysis and statistics:

-          Line 172: Describing the staining pattern is qualitative analysis. Line 174: “For this purpose” is a bit confusing, as here the description of the semi-quantitative analysis starts.

-          Line 183: Kaplan-Meier

-          Why did the authors not compare staining in primary tumors vs. metastatic lesions?

-          Did the histological pattern influence the staining?

-          Why was the staining intensity not evaluated?

-          MGT patients with bone metastasis were older than patients without. Was that effect

-          significant? If so, it should be included in the discussion.

-          Were only tumor cells included in the cell counting?

Results:

-          Table 1 and 2 need to be revised; there is no separating line between the groups, comma in the bodyweight should be replaced by point, if not rounded, the unit “kg” is already given in the table header, so it should be removed in the no metastasis group. “Canine prostate cancer patient”

-          Which cells were stained and in which cells was the staining evaluated? Only tumor cells, or osteoblasts/osteoclasts/blood vessels etc.?

-          As stated in the Methods part: some interesting data is missing regarding the different cell types, staining intensity, primary tumor vs. metastasis, influence of treatment and age… The authors stated, that they evaluated the percentage of positively stained cells, however, the results are not presented, neither as a figure, nor in the text. In this context, a comparison between primary and metastatic lesions would be of particular interest.

-          Line 221: In order to assess the role of OPN in human tumors, it would have been better to include human samples. One purpose of this study can be to compare OPN expression in canine MGT and PC with human studies, which would be part of the discussion.

-          Line 222-223: A p-value of 0.2 is not really a trend, in my opinion.

-          Figure 2: The axis legends could be revised, as the x-axis simply depicts the time and the y-axis the overall or disease-free survival [%].

Discussion:

-          Parts of the discussion are repetition from the introduction and could be summarized or moved there.

-          Another point worth being discussed is the advanced age of bone metastasis patients in the MGT group and the heterogeneous histological patterns of MGT and PC, whose influence could also be statistically evaluated.

-          As the overall goal is to establish OPN as therapeutic target, are there respective studies available in cell culture, laboratory rodents or even human patients?

The template sentence should be removed from the Author Contributions.

There are some English grammar and wording issues, I recommend to send the article to a native speaker. 

Author Response

Reviewer 4:

Analyzing osteopontin in tumor entities with frequent bone metastasis in dogs is an interesting approach and the results are indeed promising. However, some points need to be improved.

Abstract:

Canine mammary gland tumor is abbreviated CMT here, whereas in the text MGT is used.

Answer: We are sorry for this mistake. We have adjusted and standardized MGT. Thank you so much.

Introduction:

Overall, the introduction section is quite long for one analyzed marker and in some parts redundant.

Answer: we are sorry for this construction of introduction section. However, MDPI editorial office have requested us to increase the length of the manuscript and since OPN expression is not quite explored in canine tumors, it was difficult to find information. However, we agree and have excluded the redundant parts.

Line 41: Consider the wording: The metastatic cascade is not literally a physiologically coordinated process; it is rather a selection or evolvement of tumor cells, that randomly feature the mechanisms needed for all the mentioned steps of metastasis. Another example is in line 50, “radiological symptoms”. Symptoms are clinical, however, metastasis can be detected radiologically.

Answer: dear reviewer, we agree and we have excluded some parts and adjusted the introduction section. We hope this new version it will be more suitable for publication.

Lines 100-108: The passage about canine prostate cancer should be supported by some references. Furthermore, there is an issue with reference No. 17, Line 108, this reference does not state, that PC is common in dogs. This statement itself is a bit difficult, since there is still a lot research to do regarding prevalence and early stages. It would fit much better to MGT, which definitely is a common problem in veterinary medicine. I would also suggest a short sentence introducing canine MGT.

Answer: Once again, we are so sorry for this confusion. However, we have adjusted. We have used a computational program to organize the references, and an issue happened in the reference sequence. However, we have adjusted. Thank you so much.

Material and Methods:

I have some questions regarding the study design:

-          How was the sampling performed, were those biopsies, or samples obtained from mammectomies? Were the metastasis samples taken at the same timepoint as the primary tumors? In which time period did the sampling take place and in which veterinary hospital(s)? How was the follow-up information collected; were the owners contacted? Line 128: Fifth = fifty?, Line 139: type 9 error = type I error?

Answer: Dear reviewer, we collected samples from our pathology archive and in a 10 years evaluation, screening around 600 cases of dogs affected by mammary gland tumor. We tried to include the metastatic samples in this study, but we do not find sufficient tissue for inclusion of metastasis. We have excluded this information about evaluation of metastatic samples. It was our initial goal, but we were not able to include due lack of paraffin blocks at the pathology service archive. Regarding the fallow-up, they were patients that performed the complete fallow up at our veterinary teaching hospital. Therefore, we have all information in our system until death. It is important to highlight that all patients included had necropsy performed, for this reason, a low number of subject included. 

Inclusion criteria:

-          I would exclude the age from the complete clinical information, since the age is not available for some of the patients in Table 1 and 2.

Answer: Thank you so much for the suggestion. We have excluded.

-          If there is complete information about the treatment, why was it not included in the results? The choice of treatment could have a major influence on clinical outcome and survival times.

Answer: since treatment could interfere in the overall survival, we have included only patients subject only for surgery in the group of mammary gland tumor patients. In the prostate cancer group and mammary gland tumor with no possibility of surgery, only palliative treatment was performed (pain control and anti-inflammatory treatment). We have included this information in methods.

-          How was the histopathological classification performed, according to which guidelines/publications?

Answer: we are sorry that we did not include this information. For mammary gland tumor classification, we have used Zappuli et al. and for grading Peña et al. (2013). For prostate cancer, we have used Palmieri et al., (2019). We have included in methodology.

Data analysis and statistics:

-          Line 172: Describing the staining pattern is qualitative analysis. Line 174: “For this purpose” is a bit confusing, as here the description of the semi-quantitative analysis starts.

Answer: we have adjusted the staining pattern and analysis. We hope this new version it will be more suitable.

-          Line 183: Kaplan-Meier

Answer: we have adjusted.

-          Why did the authors not compare staining in primary tumors vs. metastatic lesions?

Answer: We did not have sufficient number of metastatic cases for this reason, we did not include the metastasis. The information was included in methods but was missing in the text and we excluded from methods.

-          Did the histological pattern influence the staining?

Answer: No. For exclude this hypothesis, we performed a multivariate analysis included in the manuscript showing a very weak correlation. 

-          Why was the staining intensity not evaluated?

Answer: because we did not find difference in staining intensity among the cases.

-          MGT patients with bone metastasis were older than patients without. Was that effect

-          significant? If so, it should be included in the discussion.

Answer: No, age was significant only with overall survival in our multivariate analysis.

-          Were only tumor cells included in the cell counting?

Answer: Yes, only cancer cells included in the counting for scoring the samples.

Results:

-          Table 1 and 2 need to be revised; there is no separating line between the groups, comma in the bodyweight should be replaced by point, if not rounded, the unit “kg” is already given in the table header, so it should be removed in the no metastasis group. “Canine prostate cancer patient”

Answer: we are sorry for this mistake. We have adjusted.

-          Which cells were stained and in which cells was the staining evaluated? Only tumor cells, or osteoblasts/osteoclasts/blood vessels etc.?

Answer: only neoplastic cells. We did not consider other cell subtypes.

-          As stated in the Methods part: some interesting data is missing regarding the different cell types, staining intensity, primary tumor vs. metastasis, influence of treatment and age… The authors stated, that they evaluated the percentage of positively stained cells, however, the results are not presented, neither as a figure, nor in the text. In this context, a comparison between primary and metastatic lesions would be of particular interest.

Answer: we have modified our table and presented more information including, OPN scores.

-          Line 221: In order to assess the role of OPN in human tumors, it would have been better to include human samples. One purpose of this study can be to compare OPN expression in canine MGT and PC with human studies, which would be part of the discussion.

Answer: this was a typo. We have adjusted. This is related to canine tumors. Sorry for this mistake.

-          Line 222-223: A p-value of 0.2 is not really a trend, in my opinion.

Answer: we agree and have modified the phrase.

-          Figure 2: The axis legends could be revised, as the x-axis simply depicts the time and the y-axis the overall or disease-free survival [%].

 Answer: we have adjusted as suggested.

Discussion:

-          Parts of the discussion are repetition from the introduction and could be summarized or moved there.

Answer: we have excluded several parts and modified the discussion section as suggested.

-          Another point worth being discussed is the advanced age of bone metastasis patients in the MGT group and the heterogeneous histological patterns of MGT and PC, whose influence could also be statistically evaluated.

Answer: we did a multivariate analysis to excluded this hypothesis and provided a discussion about this point was suggested.

-          As the overall goal is to establish OPN as therapeutic target, are there respective studies available in cell culture, laboratory rodents or even human patients?

Answer: we are sorry for this statement. It was very strong and we have excluded since we did not have support for this hypothesis.

The template sentence should be removed from the Author Contributions

Answer: we have excluded was suggested.

Round 2

Reviewer 1 Report

Based on the additional data and the accurate responses to the critical issues raised on the first version of the manuscript, the introductive part is more robust, and the bibliography is opportunely cited. The methodology, the English style, and the discussion section have been enhanced, too, contributing to an improvement in the quality of the work. I believe the paper can now be accepted for publication after a few minor revisions.

Minor revisions:

Page 3 line 131. Please delete “dogs, were selected 20”

Page 3 line 134. Please delete “(G-power computer program”

Page 3 line 136. Please delete “the following criteria were used:”

Even if the English stile has been imporved, I beligved that some monor changes can further improve the manuscript quality. 

Author Response

Based on the additional data and the accurate responses to the critical issues raised on the first version of the manuscript, the introductive part is more robust, and the bibliography is opportunely cited. The methodology, the English style, and the discussion section have been enhanced, too, contributing to an improvement in the quality of the work. I believe the paper can now be accepted for publication after a few minor revisions.

Answer: Thank you so much for considering our paper acceptable for publication after minor revision. Thank you again for your positive criticisms for our manuscript improvement.

Minor revisions:

Page 3 line 131. Please delete “dogs, were selected 20”

Answer: we have excluded as suggested.

Page 3 line 134. Please delete “(G-power computer program”

Answer: we have excluded as suggested.

Page 3 line 136. Please delete “the following criteria were used:”

Answer: we have excluded as suggested.

Reviewer 2 Report

Few minor comments: 

L103: studies

L130 "therefore...present study": move the sentence before (L.129)

L131: delete the repetition " , were selected 20 dogs"

L136: delete the repetition "the following criteria were used"

L155: Zappulli

L-195, 196, 199 replace d with days

L289 veterinary medicine

Only fine minor text errors

Author Response

Few minor comments: 

Answer: Thank you again for your positive criticisms for our manuscript improvement. We have modified this minor issues as requested.

L103: studies

Answer: we have adjusted as suggested.

L130 "therefore...present study": move the sentence before (L.129)

Answer: we have adjusted as suggested.

L131: delete the repetition " , were selected 20 dogs"

Answer: we have deleted.

L136: delete the repetition "the following criteria were used"

Answer: we have deleted.

L155: Zappulli

Answer: we have adjusted as suggested.

L-195, 196, 199 replace d with days

Answer: we have adjusted as suggested.

L289 veterinary medicine

Answer: we have adjusted as suggested.

Reviewer 4 Report

The adaptions made have significantly improved quality of the manuscript, so it can now be accepted. I found only one issue in Table 1: the commas in the weight column should be replaced with points, and Microsoft Word must have played tricks on the separating line between patients with other metastases and with no metastases and on the bottom line.

Author Response

The adaptions made have significantly improved quality of the manuscript, so it can now be accepted. I found only one issue in Table 1: the commas in the weight column should be replaced with points, and Microsoft Word must have played tricks on the separating line between patients with other metastases and with no metastases and on the bottom line.

Answer: We are really thankful to the reviewer for the positive comments. We have adjusted as suggested.